# Enantioselective Hydrolysis of Styrene Oxide and Benzyl Glycidyl Ether by a Variant of Epoxide Hydrolase from *Agromyces mediolanus*

**DOI:** 10.3390/md17060367

**Published:** 2019-06-20

**Authors:** Huoxi Jin, Yan Li, Qianwei Zhang, Saijun Lin, Zuisu Yang, Guofang Ding

**Affiliations:** 1Zhejiang Provincial Engineering Technology Research Center of Marine Biomedical Products, School of Food and Pharmacy, Zhejiang Ocean University, Zhoushan 316022, China; m18625623399@163.com (Y.L.); zjouzqw@163.com (Q.Z.); abc1967@126.com (Z.Y.); dinggf2007@163.com (G.D.); 2Hangzhou Institute for Food and Drug Control, Hangzhou 310019, China; saijunlin@126.com

**Keywords:** benzyl glycidyl ether, epoxide hydrolase, enantioselective, marine microorganism

## Abstract

Enantiopure epoxides are versatile synthetic intermediates for producing optically active pharmaceuticals. In an effort to provide more options for the preparation of enantiopure epoxides, a variant of the epoxide hydrolase (vEH-Am) gene from a marine microorganism *Agromyces mediolanus* was synthesized and expressed in *Escherichia coli*. Recombiant vEH-Am displayed a molecular weight of 43 kDa and showed high stability with a half-life of 51.1 h at 30 °C. The purified vEH-Am exhibited high enantioselectivity towards styrene oxide (SO) and benzyl glycidyl ether (BGE). The vEH-Am preferentially converted (*S*)-SO, leaving (*R*)-SO with the enantiomeric excess (ee) >99%. However, (*R*)-BGE was preferentially hydrolyzed by vEH-Am, resulting in (*S*)-BGE with >99% ee. To investigate the origin of regioselectivity, the interactions between vEH-Am and enantiomers of SO and BGE were analyzed by molecular docking simulation. In addition, it was observed that the yields of (*R*)-SO and (*S*)-BGE decreased with the increase of substrate concentrations. The yield of (*R*)-SO was significantly increased by adding 2% (v/v) Tween-20 or intermittent supplementation of the substrate. To our knowledge, vEH-Am displayed the highest enantioselectivity for the kinetic resolution of racemic BGE among the known EHs, suggesting promising applications of vEH-Am in the preparation of optically active BGE.

## 1. Introduction

Because of significant differences in the pharmacological activities, metabolic processes, and toxicity of enantiomers in the human body, one isomer of chiral drugs may be effective while another may be ineffective or even harmful. Chiral drugs with high enantiopurity can not only improve the specificity of the drug by eliminating side effects caused by ineffective enantiomers but also reduce the amount of the drug needed to be taken by the patient. Due to these motivations, chiral drugs have become one of the main focuses of international drug research. 

Chiral synthons play a key role in preparing chiral pharmaceuticals. As one of the important chiral synthons, chiral epoxides have been widely used to produce many chiral drugs due to their versatile reactivity. For example, chiral styrene oxide (SO) was an important chiral intermediate for the production of nematocide, levamisole, and hyperolactone C [1,2,3]. Chiral benzyl glycidyl ether (BGE) can be used for the synthesis of (+)-cryptocarya diacetate and synargentoide A, which have been found to show anti-tumor activity and have been used in the treatment of cancer pulmonary diseases [4,5].

One alternative method for preparing chiral epoxides is the enantioselective hydrolysis of racemic epoxides using epoxide hydrolase (EH, EC 3.3.2.3), which have been discovered in and cloned from many organisms such as plants, mammals, insects, bacteria, fungi, and yeasts [6,7,8]. To date, several EHs have been used to prepare chiral SO, but the enantiomeric excess (ee) or yields are always low. For example, the (*R*)-SO was obtained with an ee value of 91.6% and yield of 33.4% by using EH from *Vigna radiata* [9]. EH from *Gordonia* sp. H37 preferentially hydrolyzed (*R*)-SO, resulting in the preparation of an (*S*)-SO with ee > 99% and yield of 19.6% at 4 mM substrate concentration [10]. In addition, only a few EHs have been shown to be stereoselective for BGE. For example, EH from *bacillus alcalophilus* was used to provide (*S*)-benzyl glycidyl ether, but the ee value was only 30% [11]. EH from *Talaromyces flavus* resolved racemic BGE with moderate ee (< 98%) and selectivities (*E* < 15) [4]. The EH from *Aspergillus niger* M200 and its variants showed very low enantioselectivity (*E* < 5) for substrate BGE [12]. The limited number of EHs with high enantioselectivity for certain substrates demands studies that explore new EHs.

The ocean is rich in marine organisms that often have special metabolites and biosynthetic pathways because of their dramatically different environments. This poses potentially promising methods to screen highly for stereoselective EH from marine organisms. However, only a few EHs from marine organisms have been reported. To date, the EHs from marine microorganisms *Erythrobacter litoralis* HTCC2594, *Rhodobacterales bacterium* HTCC2654, *Erythrobacter* sp. JCS358, *Aspergillus sydowii*, and *Trichoderma* sp have been reported to produce chiral epoxides such glycidyl phenyl ether (GPE) and SO [13,14,15,16]. Xue et al. isolated a new marine microorganism *Agromyces mediolanus* from the coastal wetlands of Yancheng city, and found EH activity for epichlorohydrin (ECH), SO, and BGE [17]. However, the wild-type EH exhibited moderate enantioselectivity with only a 21.5% yield of (*S*)-ECH. Therefore, the author focused on improving enantioselectivity by site-saturation and site-directed mutagenesis of positions Ser207, Asn240 and Trp182 based on homologous modelling of the wild-type EH. Consequently, the variant (W182F/S207V/N240D) of this EH (vEH-Am) was found to enantioselectively hydrolyze racemic ECH with >99% ee and a 45.8% yield of (*S*)-ECH [18]. However, its application in the kinetic resolution of SO and BGE has still not been described. In general, the enzyme could exhibit different catalytic properties for different substrates because of the substrate specificity. 

Due to the need for more information about the catalytic properties of vEH-Am for substrates SO and BGE, we synthesized the gene of vEH-Am and expressed it in *Escherichia coli* in the present study. The recombinant vEH-Am was used in the preparation of chiral SO and BGE (Scheme 1). We also focused on improving the yield of (*R*)-SO and (*S*)-BGE by adding surfactant or intermittent feeding of the substrate.

## 2. Results and Discussion

### 2.1. Purification of the Recombinant vEH-Am

The recombinant *E. coli* BL21(DE3)/pET28a-vEH-Am was successfully constructed. After induction by 0.2 mM isopropyl β-D-thiogalactoside (IPTG), the expression of the recombinant vEH-Am was purified by His-tag affinity chromatography. The result of SDS-PAGE (Figure 1) showed a clear band with a molecular mass of approximately 43 kD in lane 2, which was consistent with the value predicted by the amino acid sequence of vEH-Am. The highest activity of vEH-Am was obtained at a pH of 8.0 and 35 °C (data not shown), which was the same as the original enzyme EH-Am [17]. This result indicated that there was no significant effect on the optimum reaction temperature and pH of the enzyme by the mutation treatment of W182F/S207V/N240D.

### 2.2. Thermal Stability of vEH-Am

Enzyme stability is a very important indicator of enzyme properties and a very important parameter in catalytic processes. Although there are many factors affecting the stability of enzymes, including inhibitors, pH, organic solvents, etc., temperature is one of the most important factors affecting the stability of enzymes. To determine the thermal stability of vEH-Am, the enzyme solutions were pre-incubated at various temperatures (30, 37, and 50 °C) for different times (0–8 h) and then cooled immediately to measure the residual activity of vEH-Am at 30 °C. As shown in Figure 2, the residual activity of vEH-Am was more than 98% after 1 h at 30 °C, while only 81% was observed at 50 °C. After incubation for 8 h, 90% of activity was still obtained at 30 °C, but only 50% at 37 °C and 30% at 50 °C. The inactivation constant and half-life of vEH-Am at each temperature were calculated based on Equations (1) and (2). The results in Table 1 suggest that the half-life of vEH-Am is 51.1 h at 30 °C. As the temperature increased, the half-life decreases sharply. The half-life decreased to 8.0 h at 37 °C, and only 4.7 h at 50 °C. Therefore, although the highest activity of vEH-Am was obtained at 35 °C, 30 °C was selected as the reaction temperature for further studies in order to avoid a significant drop in enzyme activity during the reaction process.

### 2.3. Hydrolysis of Racemic SO and BGE by vEH-Am

Enantiomerically pure epoxides can be directly prepared by the kinetic resolution of racemic epoxides with EH. However, the same enzyme shows different enantioselectivity for different substrates due to the substrate specificity. Under this context, SO and BGE were used to assess the vEH-Am’s performance and selectivity.

The time courses of the kinetic resolution of racemic SO and BGE by vEH-Am were performed under optimal reaction conditions (pH 8.0, 30 °C). Figure 3A shows that the hydrolysis rate of (S)-SO is much faster than for (R)-SO. After 10 h of reaction, (S)-SO was hydrolyzed completely, and the ee of the remaining (R)-SO exceeded 99% with a yield of 25%. For substrate BGE, as shown in Figure 3B, vEH-Am preferentially converted (R)-BGE, and the remaining epoxide (S)-BGE ee was >99% when (R)-BGE was entirely consumed, leaving 34% of (S)-BGE. These results indicate that vEH-Am can enantioselectively hydrolyze both racemic SO and BGE, but results in different enantiomers and yields. It also seemed that BGE is more of a suitable substrate than SO for vEH-Am due to its higher activity and enantioselectivity.

The kinetic resolution of racemic epoxides by EH was typically carried out in the aqueous medium, as described above. However, epoxides would spontaneously hydrolyze in this reaction system, resulting in a decrease in the yield of the optically pure epoxides. To overcome these disadvantages, the enzymatic reactions can also be performed in the presence of surfactants. Surfactants are amphoteric substances whose hydrophobic groups encapsulate hydrophobic substrates such as SO, reducing the occurrence of spontaneous hydrolysis [19]. Therefore, the hydrolysis of racemic SO and BGE by vEH-Am was investigated by adding 2% (v/v) Tween-20 to the reaction system. As shown in Figure 4, the kinetic resolution of racemic SO was also observed in the presence of Tween-20, but the hydrolysis rate of (*R*)-SO was significantly slower than that in Figure 3A (without Tween-20). After 10 h, the optically pure (*R*)-SO was obtained with a yield of 34%, which increased by 9 percentage points compared to the original 25% obtained without Tween-20. However, in BGE reactions compared to without Tween-20 (Figure 3B), the yield of (*S*)-BGE was not significantly improved by adding 2% (v/v) Tween-20 (data not shown). This result was probably attributed to the fact that the BGE was still mostly dissolved in the aqueous phase during the addition of the small amount of Tween-20.

Although kinetic resolution of racemic SO by EH has been reported in many papers, there are only a few EHs that have been obtained from marine resources. EH from the marine fish (*Mugil cephalus*) exhibited (*R*)-preferred hydrolysis activity towards racemic SO, and (*S*)-SO was obtained with >99% ee and yield of 15.4% [8]. The kinetic resolution of racemic SO also was performed by EH from a marine microorganism (*Erythrobacter* spp), (*S*)-SO with an ee > 99%, but only an 8.5% yield was obtained [14]. In our study, (*R*)-SO was obtained with >99% ee and 34% yield from racemic SO by using vEH-Am from the marine microorganism *Agromyces mediolanus*, which was better than any previous EHs from marine resources.

Optically pure BGE is an important precursor for the synthesis of many drugs and natural products. However, as shown in Table 2, there are only a few studies in the literature for the preparation of chiral BGE by EH. (*R*)-BGE with an ee of 96% and *E* value of 13 was obtained by kinetic resolution of racemic BGE with EH from *Talaromyces flavus.* The EH from *Yarrowia lipolytica* enantioselectively hydrolyzed the racemic BGE with 95% ee and an *E* value of 10.4. In addition, the EH from *Bacillus alcalophilus*, *Aspergillus niger*, *Aspergillus sydowii*, *Streptomyces griseus,* and *Trichoderma* sp. were also used to produce chiral BGE, but ee values were all very low (<60%). Of course, enzyme catalysis is affected by many conditions, so it is possible that enzymes in Table 2 could display better performance if the conditions were altered. The vEH-Am from *Agromyces mediolanus* showed the highest ee and *E* value for substrate BGE in all the reported literature, suggesting the promising future of applications for vEH-Am.

### 2.4. Kinetic Study of vEH-Am

To explore the detail of the catalytic properties of vEH-Am, kinetic studies for SO and BGE were performed using enantiopure as substrate. Kinetic parameters (*V*_m_ and *K*_m_) of vEH-Am toward (*R*)-SO, (*S*)-SO, (*R*)-BGE, and (*S*)-BGE were measured, and the results are shown in Table 3. The *V*_m_ and *K*_m_ for the (*R*)-SO were 15.9 μmol·min^−1^·mg^−1^ and 5.2 mM, whereas 3.4 μmol·min^−1^·mg^−1^ and 0.9 mM were obtained for (*S*)-SO. The lower *K*_m_ for (*S*)-SO than (*R*)-SO indicated that (*S*)-SO had a higher affinity for vEH-Am and was preferentially hydrolyzed. However, the *V*_m_ of (*S*)-SO was also lower than that of (*R*)-SO, suggesting that (*R*)-SO was hydrolyzed with a much faster rate compared to (*S*)-SO when (*S*)-SO was completely converted. The *V*_m_ for (*R*)-BGE and (*S*)-BGE were 32.1 and 63.3 μmol·min^−1^·mg^−1^, respectively, which were significantly higher than those for (*R*)-SO and (*S*)-SO. The result indicated that vEH-Am showed a higher activity for substrate BGE than SO. In addition, the (*R*)-isomer of BGE was preferentially hydrolyzed because of the lower *K*_m_, which was different from the preferential hydrolysis of (*S*)-isomer for SO. This result indicated that vEH-Am exhibits different stereoselectivity for different substrates.

### 2.5. Homology Structual Modeling and Substrate Docking

Homology modeling and molecular docking are the important methods for investigating the binding mode of the substrates and the origin of regioselectivity [9,23,24]. The 3D structure of vEH-Am was homologically modeled by Xue et al. [17,18]. In order to deeply understand the interactions between substrates and vEH-Am, the (*R*)-SO, (*S*)-SO, (*R*)-BGE, and (*S*)-BGE were antomatically docked into the active sites of vEH-Am using the Autodock 4.2 program (The Scripps Research Institute, USA). The binding modes of each enantiomer in the active site from molecular docking are shown in Figure 5. As in Figure 5A,B, both of (*R*)- and (*S*)-SO formed hydrogen bonds between Tyr 308 and the epoxide oxygen. The distance of the hydrogen bond (*d*_1_) between Tyr 308 and the epoxide oxygen in (*S*)-SO was shorter than that in (*R*)-SO. However, it was observed that the distance of the hydrogen bond (*d*_2_) between Asp181 oxygen and the attacked epoxide carbon in (*S*)-SO was slightly longer than that in (*R*)-SO. The total distance of hydrogen bonds, *d* (*d* = *d*_1_ + *d*_2_), was thought to be of particular importance for the catalytic efficacy. The *d* value should be shorter for the preferred enantiomer [25]. The differences in the modeled distance, △d, for the (*R*)-enantiomer and (*S*)-enantiomer were well consistent with regioselectivity. The *d* value for (*S*)-SO was 5.1 Å, which was shorter than 6.7 Å for (*R*)-SO, indicating that (*S*)-SO was preferably hydrolyzed by vEH-Am. For the substrate BGE (Figure 5C,D), the *d* value for (*R*)-BGE was 5.5 Å, while it was 7.3 Å for (*S*)-BGE, suggesting that (*R*)-BGE was preferentially hydrolyzed. Furthermore, the △d value between (*R*)-BGE and (*S*)-BGE was 1.8, which was longer than 1.6 Å between (*R*)-SO and (*S*)-SO, showing that vEH-Am revealed a higher regioselectivity for the kinetic resolution of racemic BGE than SO. The results were consistent with those obtained in Figure 3.

### 2.6. Biocatalytic Synthesis of (R)-SO and (S)-BGE

To test the feasibility of vEH-Am and its potential application in the synthesis of (*R*)-SO or (*S*)-BGE, the effect of the substrate concentration on the reaction was investigated. As shown in Figure 6, >99% ee of (*R*)-SO or (*S*)-BGE was obtained in all tested substrate concentrations, but the yields of them were both decreased with increasing substrate concentrations. For the substrate SO (Figure 6A), the yield of (*R*)-SO was 30.2% at the 180 μM of racemic SO, but decreased to 27.6% at concentration of 700 μM. When the concentration of racemic SO reached 2110 μM, the yield of (*R*)-SO was only 24.8%. Furthermore, a similar trend was observed for the substrate BGE (Figure 6B). With 670 μM racemic BGE as substrate, the optical purity of (*S*)-BGE was obtained with yield of 40.1%. When kinetic resolution of racemic BGE was performed at concentration of 5380 μM, the yield of (*S*)-BGE sharply decreased to 25.1%. If the concentration of racemic BGE continued to increase to 10750 μM, the yield of (*S*)-BGE was only 21.2%. The phenomenon that the yield decreased with increasing substrate concentration was also observed in kinetic resolution of epoxides by other EHs [26,27]. The result may be ascribed to enzyme inactivation caused by high concentrations of the substrate.

Substrate inhibition has always been observed in many reactions by enzyme catalysis. It is well known that the substrate inhibition can be alleviated or even eliminated by intermittent feeding of the substrate. For example, in the kinetic resolution of racemic epichlorohydrin by recombinant EH from *Agrobacterium radiobacter*, the substrate concentration increased from 320 mM to 448 mM by intermittent feeding of the substrate [28]. We therefore performed kinetic resolutions of racemic SO and BGE by intermittent feeding of the substrate to increase the yield of (*R*)-SO and (*S*)-BGE. The kinetic resolution of racemic SO was initiated at 50 mL phosphate buffer (0.2 M, pH 8.0) with 1056 μM of the initial concentration substrate and 0.5 mg of vEH-Am. A volume of 1.5 μL of racemic SO was added to the reaction system every 30 min until the final concentration was 2110 μM. As shown in Figure 7A, during the addition of the substrate, the concentration of (*R*)-SO was always higher than the concentration of (*S*)-SO, and this gap gradually increased with the extension of the reaction time. When the addition of racemic SO was halted after 2 h, the (*S*)-SO hydrolyzed rapidly, while the rate of hydrolysis of (*R*)-SO was slower. Ultimately, 701 μM of (*R*)-SO remained from 2110 μM of racemic SO after (*S*)-SO was completely consumed at 12 h of reaction. The yield of (*R*)-SO with >99% ee was 33.2%, which was significantly higher than the 24.8% obtained by adding substrate only once, as shown in Figure 6A.

The reaction of racemic BGE was performed at an initial concentration of 5380 μM, and 20 μL of racemic BGE was added every 30 min until the final concentration was 10,750 μM. Finally, (*S*)-BGE with ee > 99% was obtained at a yield of 26.9% from 10750 μM racemic BGE (Figure 7B). Compared to 21.2% obtained without addition of substrate (Figure 6B), the yield of (*S*)-BGE was significantly improved by intermittent feedings. However, it was observed that the extent of improvement in yield of (*S*)-BGE (increasing 5.7%) was not as significant as (*R*)-SO (increasing by 8.4%). This similar result was also observed in biosynthesis of (*R*)-SO and (*S*)-BGE by addition of Tween-20, as shown in Figure 4. These results indicate that the substrate inhibition of SO is stronger than that of BGE, which may be mainly attributed to the more hydrophobic tendency of SO than BGE. Another promising way to overcome substrate inhibition is to perform the reaction in an aqueous/organic two-phase system, which had been used in previous research to improve substrate concentration [28,29]. In the future, we would try to use a two-phase reaction system to increase the substrate concentration and yield of the chiral epoxide.

## 3. Materials and Methods

### 3.1. Materials

(*R*,*S*)-SO, (*R*)-SO, (*S*)-SO, (*R*,*S*)-BGE, (*R*)-BGE, and (*S*)-BGE were obtained from Aladdin Bio-Chem Technology Co., Ltd. (Shanghai, China). *E. coli* BL21(DE3) and pET-28a were used for the expression of vEH-Am. All other chemicals were of analytical grade from commercial sources.

### 3.2. Cell Culture and Protein Expression

The vEH-Am sequence was synthesized according to the gene of EH from *Agromyces mediolanus* ZJB120203 (GenBank accession no. JX467176) and mutant sites (W182F/S207V/N240D) using the PCR method [30]. Six His-tags were added at the 3′ end of the vEH-Am gene. The synthesized gene was inserted into pET28a between NcoI and XhoI sites. The recombinant *E. coli* was obtained by transforming the pET28a-vEH-Am into *E. coli* BL21(DE3). The strain was inoculated into 50 mL of LB liquid medium containing 50 μg/mL kanamycin and cultured overnight at 37 °C as a seed solution. A volume of 0.5 mL of seed solution was then transferred into 50 mL of the same LB medium and incubated at 37 °C at 200 rpm until the OD_600_ of the fermentation liquid was about 0.7. Isopropyl β-D-thiogalactoside (IPTG) was added with the final concentration of 0.2 mM. The fermentation liquid was further cultured for 8 h at 28 °C and then centrifuged at 10,000 rpm for 15 min to obtain cells.

### 3.3. Purification of vEH-Am

Four grams of wet cells were suspended in 40 mL of phosphate buffer (20 mM, pH 8.0) and broken by a 30 min ultrasound treatment. The disrupted solution was centrifuged at 10,000 rpm for 20 min to remove cell debris. The supernatant solution (crude enzyme solution) was collected for subsequent separation and purification. The purification column was Nickel–nitrilotriacetic (Ni-NTA), and the packed volume was 10 mL. The Ni-NTA column was equilibrated with buffer A (20 mM phosphate buffer, 500 mM NaCl and 20 mM imidazole, pH 8.0) for 10 min. The crude enzyme solution was then inflowed into the column at a rate of 1 mL/min. After the unbound proteins were eluted by buffer A, the target protein vEH-Am was collected by elution at a rate of 3 mL/min with buffer B (20 mM phosphate buffer, 500 mM NaCl and 500 mM imidazole, pH 8.0). The enzyme solution was dialyzed overnight in a phosphate buffer of 20 mM (pH 8.0), and the purified enzyme was analyzed by 12% sodium dodecyl sulfate polyacrylamide gel electrophoresis (SDS-PAGE).

### 3.4. Thermostability of vEH-Am

The temperature stability of vEH-Am was determined in 10 ml of phosphate buffer (0.2 mM, pH 8.0) containing 0.1 mg of vEH-Am. The mixture was pre-incubated for different times at 30, 37, and 50 °C, and the remaining activity was assayed by adding 10 μL of racemic BGE. The inactivation constant *k_d_* and half-life t_1/2_ were calculated based on the following formula:
(1)$$\ln(\frac{\left[E\right]}{\left[E_{0}\right]})=-k_{d}\times t$$
(2)
t_1/2_ = 0.693/*k_d_*

### 3.5. Activity Assay and Analytical Methods

The racemic substrate (1 μL SO or 10 μL BGE) and 0.1 mg vEH-Am were mixed in 10 mL of phosphate buffer (0.2 M, pH 8.0). The reaction was performed for 15 min at 30 °C at 150 rpm. The 1.0 mL reaction solution was then taken and added to 2 mL of hexane. After 3 min, 1 mL of the organic layer was separated and dried with anhydrous sodium sulfate. The reaction solution was treated with a 0.2 μm filter membrane after centrifugation at 4000 rpm for 3 min and then analyzed by HPLC to determine the enzyme activity and ee value. The concentrations of the substrates were analyzed using an Agilent LC 1260 with a CHIRALPAK AS-H column (5 μm, 4.6 × 250 mm). Detection conditions were as follows: 20 μL injection volume, mobile phase n-hexane/isopropanol (95:5), 210 nm detection wavelength, 30 °C column temperature, 1 mL/min flow rate. Retention times: 5.3 min for (*R*)-SO, 5.6 min for (*S*)-SO, 8.3 min for (*S*)-BGE, and 9.8 min for (*R*)-BGE. The ee was calculated from the concentrations of the two enantiomers based on the formula: ee (%) = (*S* − *R*)/(*S* + *R*) × 100. The enantiomeric ratio (*E* value) was calculated based on the ee of the remaining epoxide and the conversion (*C*) of racemate according to equation: *E* = ln[(1 − *C*)(1 − ee)]/ln[(1 − *C*)(1 + ee)]. One unit of enzyme activity was defined as the amount of enzyme required to convert 1 μmol of substrates at 30 °C.

### 3.6. Determination of Kinetic Properties

The kinetic study of the vEH-Am was performed by measuring the initial rate at different concentrations of (R)-SO, (S)-SO, (R)-BGE, and (S)-BGE. The kinetic parameters V_m_ (maximum reaction rate) and K_m_ (Michaelis constant) were calculated based on the Michaelis–Menten equation: 1/v = K_m_/V_m_[S] + 1/V_m_, where v is the initial velocity and [S] is the substrate concentration.

### 3.7. Homology Modeling and Docking

The three-dimensional structure of vEH-Am was generated by Modeller 9.12 in Discovery studio (DS) 2.1 (Accelrys Software, San Diego, CA, USA) based on the crystal structure of EH (PDB accession no. 4i19) [18]. The docking studies were performed by Autodock 4.2 (The Scripps Research Institute, USA).

### 3.8. vEH-Am Hydrolysis of Racemic SO and BGE

Hydrolysis of racemic SO and BGE was performed in 50 mL 0.2 M sodium phosphate buffer (pH 8.0) containing 0.5 mg vEH-Am. The reactions were initiated by adding various concentrations of racemic substrates at 30 °C and 200 rpm. The 1.0 mL mixtures were taken out regularly at different times intervals and added to 2 mL of hexane to stop the reaction. The hydrolysis progressions of racemic SO and BGE were analyzed by measuring the concentrations of each enantiomer.

## 4. Conclusions

In this study, we report on the kinetic resolution of racemic SO and BGE by the particular variant of epoxide hydrolase from *Agromyces mediolanus* (vEH-Am). The vEH-Am enantioselectively hydrolyzed (*S*)-SO and (*R*)-BGE, leaving (*R*)-SO and (*S*)-BGE with >99% ee. Moreover, the yield of optically pure (*S*)-BGE reached 34%, which was the highest among all reported epoxide hydrolases. Molecular docking simulations showed that the hydrogen bonds between vEH-Am and (*S*)-SO or (*R*)-BGE were shorter than the other enantiomers. The concentration of the substrate had a negative impact on the yield of (*R*)-SO or (*S*)-BGE, which could be significantly improved by adding Tween-20 or intermittent feeding of the substrate. This study lays the theoretical foundation for the application of vEH-Am in the preparation of enantiopure SO and BGE.

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
