# Peer review of "Enantioselective Hydrolysis of Styrene Oxide and Benzyl Glycidyl Ether by a Variant of Epoxide Hydrolase from Agromyces mediolanus"

_marinedrugs, 2019, doi:10.3390/md17060367_

Round 1

Reviewer 1 Report

This manuscript by Jin et al. reports the enantioselective hydrolysis of SO and BGE with the engineered epoxide hydrolase vEH-Am (W182F/S207V/N240D) from Agromyces mediolanus. The reaction parameters such as pH, temperature, concentration, and reaction medium were optimized. The docking study was carried out to address the origin of the kinetic resolution by vEH-Am. But, the epoxide hydrolase activity of AmEH (ZJB120203) and its variants including vEH-Am previously has been reported by Zheng et al. This work just extended vEH-Am to the enantioselective hydrolysis of SO and BGE. Thus, it seems to this reviewer that this work lucks novelty, although the enantioselectivity of vEH-Am is superior rather than other EHs from marine resources. Also, to accomplish the aim of this work exploring the detail of the catalytic properties of vEH-Am and substrate specifity as the author mentioned, the quantitative discussion on the basis of the kinetic study and further substrate scope and limitation (e.g. substituted SOs) would be required. The chiral HPLC profile for the hydrolase catalytic reactions should be provided in a supplemental data. 

Author Response

    Dear reviewer,

    Thanks for the valuable comments. We have revised the manuscript according to the comments that we received. The modified part of the text has been marked in red. We hope the revised manuscript and response will meet your standard, thanks.

    Response to the comments: 

    Thanks for your comments. We are very honored to receive your professional comments. Although the enzyme vEH-Am previously had been reported, we are the first to apply this enzyme to the synthesis of chiral BGE with the highest enantioselectivity in known reports. Furthemore, according to the comments of reviewer, the kinetic parameters for (R)-SO, (S)-SO, (R)-BGE, and (S)-SO were determined and supplemented in the article for exploring the detail of the catalytic properties of vEH-Am. In addition, we have found in the preliminary experiment that vEH-Am show moderate enantioselectivity for p-nitrostyrene oxide and p-chlorostyrene oxide, and low enantioselectivity for 1,2-epoxyoctane. Based on the higher enantioselectivity for SO and BGE, the catalytic properties of vEH-Am toward SO and BGE were deeply studied in this research. Of course, more different substrates will be performed in the future. Finally, the chiral HPLC spectrum for the hydrolase catalytic reactions had been provided in the supplemental data. I sincerely hope that you can give me a chance to publish this paper. Thanks again for your professional suggestion.

Reviewer 2 Report

The authors of this manuscript discovered and characterized a novel variant of epoxide hydrolase that has the potential to be utilized in the preparation of pharmacological benzyl glycidyl ether.  Given the high enantioselectivity of the described enzyme, this research provides crucial information regarding optimal parameters, which are required to obtain enantiopure styrene oxide and benzyl glycidyl ether. Assuming that authors will address the comments below, I highly recommend this manuscript for publication.
I have three main comments/questions:
Line 64: the authors mention a novel variant of a known enzyme which contains now several mutations. It would be useful to describe here in a few words how and why these mutations were previously engineered.
Figure 1, lane 1. Does it show the whole cell extract? Why are only three bands present then?
Line 158: Do other studies also consider possible substrate inhibition in their reactions? Can, for instance, an enzyme from Trichoderma sp. likely have higher than 60 % ee in case of altered substrate concentrations?  Perhaps, the authors could mention that enzymes discussed in Table 2 could also display better performance in the preparation of the chiral BGE if the conditions are altered.

Author Response

     Dear reviewer,

    Thanks for the valuable comments. We have revised the manuscript according to the comments that we received. The modified part of the text has been marked in red. We hope the revised manuscript and response will meet your standard, thanks.

    A point-by-point response to the reviewer’s comments as follows:

    Question 1: Line 64: the authors mention a novel variant of a known enzyme which contains now several mutations. It would be useful to describe here in a few words how and why these mutations were previously engineered.

    Answer: According to your comments, we have described in a few words how and why these mutations were previously engineered in paper. Thanks.

    Question 2: Figure 1, lane 1. Does it show the whole cell extract? Why are only three bands present then?

    Answer: Thanks for your professional review. In fact, there are many small protein bands in lane 1, but they are less significant because of lower protein concentrations. This result is mainly due to the low concentration of cells in the cell suspension during disruption.

    Question 3: Line 158: Do other studies also consider possible substrate inhibition in their reactions? Can, for instance, an enzyme from Trichoderma sp. likely have higher than 60 % ee in case of altered substrate concentrations?  Perhaps, the authors could mention that enzymes discussed in Table 2 could also display better performance in the preparation of the chiral BGE if the conditions are altered.

    Answer: Thanks for your professional comments. The effect of substrate concentration on kinetic resolution was not available in other studies. Therefore, it cannot be concluded whether there is substrate inhibition in their reactions. The comparisons between vEH-Am and other enzymes in Table 2 are based on the data reported in their literatures. Theoretically, it is possible that enzymes discussed in Table 2 could display better performance in the preparation of the chiral BGE if the conditions are altered. According to your comments, we had mentioned it in text, thanks.

Round 2

Reviewer 1 Report

Current version seems to be acceptable for Marine Drugs.